# Lamiaceae in Mexican Species, a Great but Scarcely Explored Source of Secondary Metabolites with Potential Pharmacological Effects in Pain Relief

**DOI:** 10.3390/molecules26247632

**Published:** 2021-12-16

**Authors:** Alberto Hernandez-Leon, Gabriel Fernando Moreno-Pérez, Martha Martínez-Gordillo, Eva Aguirre-Hernández, María Guadalupe Valle-Dorado, María Irene Díaz-Reval, María Eva González-Trujano, Francisco Pellicer

**Affiliations:** 1Laboratorio de Neurofarmacología de Productos Naturales, Dirección de Investigaciones en Neurociencias, Instituto Nacional de Psiquiatría Ramón de la Fuente Muñiz, Mexico City 14370, Mexico; albertoh-leon@imp.edu.mx (A.H.-L.); mmspoodles@gmail.com (G.F.M.-P.); pellicer@imp.edu.mx (F.P.); 2Programa de Posgrado en Ciencias Biológicas, Facultad de Medicina, Universidad Autónoma de México, Ciudad Universitaria, Mexico City 04510, Mexico; 3Herbario de la Facultad de Ciencias, Departamento de Biología Comparada, Facultad de Ciencias, Universidad Nacional Autónoma de México, Ciudad Universitaria, Mexico City 04510, Mexico; mjmg@ciencias.unam.mx; 4Laboratorio de Productos Naturales, Departamento de Ecología y Recursos Naturales, Facultad de Ciencias, Universidad Nacional Autónoma de México, Ciudad Universitaria, Mexico City 04510, Mexico; eva_aguirre@ciencias.unam.mx; 5Departamento de Farmacología, Facultad de Medicina, Universidad Nacional Autónoma de México, Ciudad Universitaria, Mexico City 04510, Mexico; lvalle_59@hotmail.com; 6Centro Universitario de Investigaciones Biomédicas, Universidad de Colima, Colima 28045, Mexico; idiazre@ucol.mx

**Keywords:** Lamiaceae, *Salvia*, *Agastache*, pain, nociception, inflammation

## Abstract

The search for molecules that contribute to the relief of pain is a field of research in constant development. Lamiaceae is one of the most recognized families world-wide for its use in traditional medicine to treat diseases that include pain and inflammation. Mexico can be considered one of the most important centers of diversification, and due to the high endemism of this family, it is crucial for the in situ conservation of this family. Information about the most common genera and species found in this country and their uses in folk medicine are scarcely reported in the literature. After an extensive inspection in bibliographic databases, mainly Sciencedirect, Pubmed and Springer, almost 1200 articles describing aspects of Lamiaceae were found; however, 217 articles were selected because they recognize the Mexican genera and species with antinociceptive and/or anti-inflammatory potential to relieve pain, such as *Salvia* and *Agastache*. The bioactive constituents of these genera were mainly terpenes (volatile and non-volatile) and phenolic compounds such as flavonoids (glycosides and aglycone). The aim of this review is to analyze important aspects of Mexican genera of Lamiaceae, scarcely explored as a potential source of secondary metabolites responsible for the analgesic and anti-inflammatory properties of these species. In addition, we point out the possible mechanisms of action involved and the modulatory pathways investigated in different experimental models. As a result of this review, it is important to mention that scarce information has been reported regarding species of this family from Mexican genera. In fact, despite *Calosphace* being one of the largest subgenera of *Salvia* in the world, found mainly in Mexico, it has been barely investigated regarding its potential biological activities and recognized bioactive constituents. The scientific evidence regarding the different bioactive constituents found in species of Lamiaceae demonstrates that several species require further investigation in preclinical studies, and of course also in controlled clinical trials evaluating the efficacy and safety of these natural products to support their therapeutic potential in pain relief and/or inflammation, among other health conditions. Since Mexico is one of the most important centers of diversification, and due to the high endemism of species of this family, it is crucial their rescue, in situ conservation, and investigation of their health benefits.

## 1. Introduction

The term Labiatae comes from the Latin word “labia”, which means “lip”, and refers to the peculiar morphological characteristic of all the species that belong to this family, which have the corolla split into an upper lip and a lower one. This term precedes the name Lamiaceae, which comes from the Greek “laimos” referring to the “gaping mounth” of the corolla [1]. The Lamiaceae family belongs to the order Lamiales, in the clade Lamids, in the Eudicots [2]. It is the sixth largest family of angiosperms comprising 12 subfamilies, 16 tribes, 9 subtribes, 236 genera, and over 7173 species [1,3]. A wide range of substances isolated from plants belonging to this family produce antibacterial, cytotoxic, antioxidant, anti-inflammatory and insecticidal activities [4]. Various studies report that the members of this family are a source of phytochemical compounds with health benefits or play an active role in the improvement of diseases, mainly due to the content and type of compounds, the main ones being essential oils, terpenoids, phenolic acids and flavonoids [5,6], many of which can be used to relieve pain.

It is well documented that healing with medicinal plants is as old as humanity itself, perhaps mainly due to pain. Hippocrates (5th century BC) was the first Greek writer to use the word analgesia in a medical rather than a philosophical context, as well as derivative words related to pain when using plants to relieve it. Dioscorides (1st century BC) was a Greek philosopher who explored the therapeutic properties of plants, including those used for pain relief, describing the narcotic effects of the plant mandragora. As a military physician and pharmacognosist, Dioscorides differentiated among a few species from the genus *Mentha* (Lamiaceae), which were grown and used to relieve headache and stomachache. Hey used expressive and precise adjectives and well-defined characteristics of pain, such as location, duration, or relation to other symptoms, to elucidate a disease process [7].

Nowadays, according to the International Association for the Study of Pain, “pain” has been defined as an unpleasant sensory and emotional experience associated with, or resembling experiences associated with, actual or potential tissue damage [8]. It can be classified as functional (nociceptive or inflammatory) [9] or dysfunctional (neurogenic, neuropathic and psychogenic) because of its origin or etiology [10], or as acute or chronic because of its temporality [9]. In acute pain, nociceptors are activated in the site of tissue damage, while chronic pain is commonly triggered by injury or an illness, and it can be perpetuated by factors other than the cause of pain [10]. Acute pain associated with tissue damage can last for less than 1 month, but sometimes it can last for more than 6 months, at which point it becomes chronic pain. Preclinical studies have shown that neuronal expression of new genes (the basis for neuronal sensitization and remodeling) occur within 20 min of injury. Chronic pain can cause long-term behavior and histological changes within approximately one day after interventions, such as transient nerve ligation [8]. Among the characteristics that commonly occur in patients with chronic and dysfunctional pain are hyperalgesia (exacerbated sensitivity to painful stimuli), allodynia (painful response to harmless stimuli) and hyperesthesia (abnormal sensitivity to sensory stimuli) [11].

Nociception is not the same as the term pain, it is the mechanism through which harmful stimuli are transmitted to the central nervous system (CNS). This term is used in the preclinical evaluation of the plants or bioactive constituents. Nociceptors are neurons sensitive to noxious stimuli and are in the skin, blood vessels, muscles, fascia, joints and viscera. They are predominantly myelinated (A-δ) or unmyelinated (c) fibers, activated by noxious stimuli (mechanical, thermal, cold and chemical), which carry these signals to the CNS [12,13,14]. All tissues, with the exception of the CNS neuropil, are innervated by afferent fibers, although their properties differ markedly, depending on whether they are somatic (skin, joints, muscles) or visceral fibers (cardiovascular or respiratory tissue, gastrointestinal or renal tract), and reproductive systems [15].

Pathophysiological mediators of pain and inflammation are generated by several important sources. These mediators can act through a multiplicity of receptors that are widely distributed in central and peripheral nerves and coupled to heterotrimeric G proteins, as occurs in opioids and 5-HT_1A_ inhibitory receptors associated with multiple second messengers formation (cAMP, cGMP, DAG, IP3, intracellular Ca^+2^, NO) and protein kinases (A, C or G) to promote the phosphorylation of multiple targets [10]. Other pharmacological receptors involve activity through ion channels, e.g., excitatory amino acids or acetylcholine (which activates the nicotinic receptor), to control the ionic permeability of the membrane and muscle contraction [15,16].

In the case of inflammation, it involves an integration of several inflammatory mediators, such as prostaglandin E2, bradykinin, substance P, histamine, adenosine and serotonin, sensitizing nociceptors after mechanical and thermal stimuli. It is commonly reported that mediators in an inflammatory condition are cytokines. In this respect, Toll-like receptors (TLRs) activate proinflammatory cytokine profiles in macrophages, altering the homeostatic regulation of the immune system. Macrophages are essential components of the innate and adaptive immune systems, and therefore play a central role in inflammation, host defense, and tissue repair [17,18]. Depending on the microenvironment, these cells are functionally classified into two main types: classically activated proinflammatory M1 macrophages and alternately activated M2 macrophages. M1 macrophages are induced by Th1 cytokines such as interferon γ (IFNγ) and tumor necrosis factor α (TNF-α) or by lipopolysaccharide (LPS) and typically attack microorganisms and tumor cells, and express inducible nitric oxide synthase (iNOS) and most of the TLRs [18]. In contrast, M2 macrophages are induced by Th2 cytokines such as interleukins (ILs)-4, -13, and -10 and transforming growth factor β (TGF-β). Könner and Brüning [19] demonstrated that TLR2 and TLR4 are closely related to the systemic inflammatory response. TLRs (of which there are 10 types in humans and 12 in mice), contain adapter proteins, the recruitment of which is followed by a signaling pathway that activates nuclear factor kappa B (NF-kB), activator protein 1 (AP-1), signal transducer and activator of transcription 1 (STAT-1) and interferon regulatory factor (IRF), which mediate inflammation as well as cytokines release [20,21]. NF-kB is an important nuclear transcription factor in the regulation of the inflammatory response. It participates in biological processes that involve inflammation, immunity, differentiation, cell growth, tumorigenesis and apoptosis [22]. NF-kB is regulated by binding to inhibitory molecules such as the nuclear factor of kappa light polypeptide gene enhancer in B-cells inhibitor, alpha (IkBα). The NF-kB p65 subunits dissociate from their inhibitory protein IkBα by translocating from the cytoplasm to the nucleus where they influence the expression of proinflammatory cytokines such as TNF-α, IL-1β, IL-6 and IL-8 [23]. Therefore, the prevention of nuclear translocation of NF-kB may work as a potential therapeutic target. The transcription factor Nrf2 is largely responsible for the inducible expression of proteins involved in the response to oxidative stress, cell protection and the inhibition of the expression of inflammatory cytokines, such as IL-6 and IL-1β. Furthermore, Nrf2 is associated with the NF-kB-mediated transcription of proinflammatory cytokines genes [24]. The signaling pathway of mitogen-activated protein kinases (MAPKs) consists of a family of serine/threonine kinases that are activated by several stimuli, including different inflammatory factors [25]. MAPK proteins control fundamental cellular processes, such as proliferation, differentiation, metabolism, inflammation and apoptosis. As MAPK and NF-kB can synergistically collaborate to induce proinflammatory cytokines [25,26], secondary metabolites of phytopharmaceuticals with inhibitory capacity of NF-kB and MAPK may have potential therapeutic advantages in the treatment of inflammatory diseases. It has been demonstrated that many plant constituents can interact with one or more than one of those previously mentioned biological targets, which are involved in producing, enhancing or relieving pain alone, or are associated with inflammation because of tissular damage.

Several aspects of Lamiaceae are addressed in this review, emphasizing that Mexican genera have not been explored enough as source of medicinal species. We provide evidence of the minimal number of species explored to investigate potential secondary metabolites responsible for their antinociceptive and anti-inflammatory properties and point out the mechanisms of action involved and modulatory pathways by using different experimental models. The integration of relevant information on some species belonging to Mexican genera of Lamiaceae and substances derived from these and other investigated plants reinforces the evidence already reported regarding various species from different regions around the world on their medicinal properties as promising alternatives for potential analgesics and anti-inflammatory drugs.

## 2. Results

### 2.1. Lamiaceae Description

The species belonging to this family are generally characterized by being annual or perennial herbaceous plants or shrubs, sub-shrubs and less commonly trees or vines; occasionally with stolons or rhizomes; often with aromatic oils; stems being erect or prostrate, generally tetragonal, due to the presence of large bundles of collenchyma, with no spines [27], and with or without a glandular trichome indument. Opposite leaves, generally decussate, sometimes whorled, simple or less frequently compound (*Vitex*), dentate or crenate; the petiole being present or absent; and stipules being absent [28].

Terminal or axillary inflorescences, thyrsoids, are usually found with cymes or whorls arranged in spikes, racemes, panicles or a capitulum, and bracts are usually present, being persistent or deciduous. The flowers are usually bisexual, hypogynous, zygomorphic, and rarely actinomorphic; have a persistent, gamosepalous, tubular to widely campanulate calyx; have four to five (to nine) lobes, which are imbricated; have a gamopetalous corolla, generally with five lobes, equal or sub-equal, frequently bilabiate, and in that case the upper lip bilobed and the lower lip are trilobed, with imbricated lobes, and a short or long tube; have four stamens, which are didynamous, rarely equal, sometimes reduced to two and sometimes with staminodia present, and epipetalous; generally have free filaments; have anthers longitudinally dehiscent; have a hypogynous disc, which usually fleshy, and sometimes divided into four glands; have a bicarpellary gynoecium, which is usually tetralocular due to a false septum, upper ovary, one style, which is gynobasic, and less frequently terminal, and a filiform, usually with two stigmatic lobes, equal or unequal; have four ovules, one per locule, erect, and the fruit is tetralobed and, indehiscent, usually with four nuts, which are dry, smooth or slightly tuberculated or reticulated-rough. The seeds are usually number four [3,27,28]. Photographs providing examples of species of Lamiaceae from Mexican Salvias (*Calosphace* subgenus) are shown in Figure 1.

### 2.2. Geographical Distribution

In various regions worldwide, members of the Lamiaceae can be found. This family contains 236 genera and approximately 7173 species, growing in areas with tropical and temperate climates from 0 to 2500 m above sea level [3]. Several species are found to be abundant in mountainous areas, with a temperate climate, although it is possible to find the *Hyptis* and *Asterohyptis* genera in dry and hot areas [28,29]. There are six regions of high diversity in the world [27,30]: the Mediterranean and Central Asia [31], Africa and Madagascar [32], China [33], Australia, South America [34] and North America (including Mexico) [28,35] (See Figure 2A).

In Mexico, Lamiaceae is the family with the eighth greatest diversity and the number of species of this family in Mexico represents 5.5% of the species belonging to this family worldwide. For that reason, Mexico can be considered one of the most important centers of diversification, and due to the high endemism of this family, it is crucial for the in situ conservation of this family [35] (Figure 2B).

### 2.3. Lamiaceae and Some of Its Genera in Mexico

Lamiaceae is one of the most diverse families in Mexico, after only Asteraceae, Fabaceae, Poaceae, Orchidaceae, Cactaceae, Euphorbiaceae, and Rubiaceae, representing 13.55% of the genera and 8.23% of the world’s species, with an endemism of 66.2% [35]. Mexico has 31 genera and 598 species. The most diverse genus is *Salvia*, with 306 species, where the subgenus *Calosphace* is one of the most diverse, with 295 species from the 580 included in the group [36]. Oaxaca is the state with the greatest diversity, while Jalisco houses the largest number of endemic species. Even though no genus of the family is endemic in the country, plants of the *Cunila* or *Hedeoma* genera have an endemism of up to 60% [35].

According to Harley et al. [3], from the 236 genera of Lamiaceae, 226 were assigned to seven subfamilies: *Ajugoideae*, *Lamioideae*, *Nepetoideae*, *Prostantheroideae*, *Scutellarioideae*, *Symphorematoideae*, and *Viticoideae*. *Cymarioideae*, *Peronematoideae*, and *Premnoideae* were added later [37]. From these, six are found in Mexico. Finally, Callicarpa and Tectona were transferred from the *Verbenaceae* family being recognized as independent subfamilies latterly [31] (See Figure 3).

### 2.4. Pain and Some of Mexican Lamiaceae Genera to Alleviate It

Currently, a wide variety of herbs are found in the markets of several cities around the world [38], where the Lamiaceae family contains species of economic value because of its culinary or flavoring and medicinal properties. Species from this family have been widely used since ancient times, as spices and herbal teas in traditional medicine. Velongiarious members of this family are specifically used as a source of essential oils [27]. In fact, a variety of healing properties is attributed to each plant from Lamiaceae [29]. Several of them are repeatedly recommended to treat the same disease, suggesting that similar constituents are included in them [39] (Table 1). Despite this, reports describing pharmacological evidence on the antinociceptive and anti-inflammatory effects, bioactive compounds isolated and mechanism of action from Mexican Lamiaceae species are scarce in the literature. Most of the investigations have evaluated polar extracts from nature exploring the following genera: *Hyptis*, *Lavandula*, *Leonurus*, *Melissa*, *Marrubium*, *Mentha*, *Ocimum*, *Origanum*, *Sage*, *Satureja*, *Stachys*, *Scutellaria*, *Sideritis*, and *Teucrium* [40]. Antinociceptive and anti-inflammatory effects were observed in polar extracts of *Marrubium* evaluated in formalin and carrageenan tests [41,42]. The analgesic-like response of the ethanol and hydroalcoholic extracts of *Hyptis* were evaluated in the writhing, formalin, tail immersion, carrageenan and hyperalgesia induced by glutamate or capsaicin tests in mice [43,44]. In a similar manner, the hydroalcoholic and aqueous extracts of species from *Teucrium* [45,46,47] or *Scutellaria* [48] have shown these activities. Analysis of hydroalcoholic extracts of species from the *Stachys* genus was carried out in mice using formalin, acetic acid-induced writhing, and light tail-flick tests [39], and that of *Leonurus cardiaca* L. was performed by using formalin, tail-flick, and hot-plate tests in mice [49].

In this manuscript, there is a special interest in plants belonging to this family, since they are widely and traditionally used in Mexican folk medicine for the relief of pain and/or inflammation, as are some species of the genera *Salvia* and *Agastache*.

### 2.5. Salvia Species Used in Pain Relief

*Salvia*’s name comes from the Latin *salvare*, which means to save and heal. One of the most ancient species of this genus is *Salvia officinalis* L. despite its Mediterranean origin, it is also of great medicinal use in Mexico, and is commonly known as sage. It was used by Egyptians, Greeks, and Romans to treat ulcerations. Nowadays, its anti-inflammatory properties are useful for the treatment of buccal conditions such as amigdalitis, faringitis and gingivitis, which might be because of the bioactive effect of components such as ursolic acid [91]. Its anti-inflammatory and antioxidant properties are influenced by the cytokine’s mediators considering its marked properties that inhibit the increase in IL-33 and TNF-α levels and the amplification of NF-kB expression and its activation [92].

Another species of Mediterranean origin with great medicinal utility to relieve pain in Mexico is *Rosmarinus officinalis* L., a Lamiaceae species recently reclassified as *Salvia Rosmarinus* Spenn. [40]. This species, cultivated in Mexico, possesses a broad spectrum of antinociceptive activities already evidenced in several acute and chronic experimental models, such as the writhing, formalin and gout arthritic-like pain tests, by preparing polar and non-polar extracts [93,94]. The responsible bioactive compounds of this species are flavonoids and triterpenoids, including hesperidin and ursolic, oleanolic, and micromeric acids [94]. Their mechanisms of action involve calcium channels and central inhibitory receptors or peripheral actions depending on the kind of pain induced [94,95]. The involvement of several mechanisms of action allowed the researchers to obtain a synergistic antinociceptive response in the presence of clinical analgesics and other medicinal plants used to alleviate pain [96].

Scarce information was found in the literature describing *Salvia* and *Agastache* genera distributed in Mexico and used for pain reliefDue to this, our group explored some of these genera to obtain pharmacological evidence of their medicinal properties, for example the cases of *S. divinorum* Epling & Jativa, S. *semiatrata* Zucc. and *S. amarissima* Ortega, as well as their main constituents as promising anti-inflammatory, antioxidant and antinociceptive agents, highlighting their mode of action in experimental models of pain as follows:

*Salvia divinorum* (1962) is a member of the Sage family that has been historically used for divination and shamanism by the Mazatecs from Oaxaca, Mexico, because of this hallucinogenic properties with a short duration. However, its leaf extracts have also been reported as useful medicinal species to relieve pain [97]. It has shown a wide spectrum of antinociceptive activities in preclinical studies using acute nociception in abdominal pain and in the neurogenic and inflammatory test of formalin [98] but also in chronic pain models, such as in neuropathic pain involving electroencephalographic changes [99]. Its bioactive compounds are from salvinorin, mainly salvinorin A, a neoclerodane diterpene, by involving kappa opioid, 5-HT_1A_ serotonin and CB1 cannabinoid receptors as the main mechanisms of action [99,100,101].

*Salvia amarissima* is another endemic species in Mexico used in traditional medicine to treat disorders attributed to a cold state such as anxiety in the CNS, as well as gastrointestinal ailments and pain relief. It has been evaluated using several preclinical animal models of pain preparing different kinds of extracts, where medium polar and polar extracts have been the most bioactive [102]. It has also been observed that medium polar constituents are involved in the inhibition of neurogenic and inflammatory nociceptive responses through the participation of opioid and TRPV1 and 5-HT_1A_ serotonin receptors [103]. The presence of a neoclerodane terpene named amarisolida A and the flavonoid pedalitin have been implicated in the bioactive and abundant constituents [102]. Their properties have also been reported in metabolic alterations such as diabetes, since they produce a significant antihyperglycemic action in vivo during an oral sucrose tolerance test by an alfa-glucosidase inhibitory activity [104]. The presence of flavonoids and terpenoid bioactive constituents has also been associated with another enzymatic and regulatory protein inhibition such as in protein tyrosine phosphatase 1B (PTP-1B), where different kinds of amarisolide have been characterized [105], as well as flavonoids such as rutin, isoquercitrin and pedalitin already associated with anxiolytic and/or antinociceptive activities [106,107]. Diterpenes derived from *S. amarissima* also possess modulatory capability in protein multidrug resistance and cytotoxic activity [108]. All these properties together suggest their useful application in the comorbidity of diabetes and pain such as in neuropathic pain or even in cancer.

*Salvia semiatrata* is a species used as a tranquilizer and to relieve pain in folk medicine in Santiago Huauclilla, Oaxaca, Mexico. Preclinical evidence was recently reported regarding its significant effects as an anxiolytic and antinociceptive in several experimental models in which the presence of the neo-clerodane diterpene 7-keto-neoclerodan-3,13-dien-18,19:15,16-diolide was identified as being partially responsible [109]. Pain is as strongly associated with anxiety as with depressive disorders, this comorbility can exacerbate the other significantly [110]. This kind of comorbidity can be modulated by the dual activity of herbal therapy, as it was observed in anxiolytic and antinociceptive effects of *S. semiatrata* using similar doses [109].

### 2.6. Agastache Species to Alleviate Pain and Inflammation

*Agastache mexicana* (Kunth) Lint & Epling is a plant in high demand that has long been used in Mexican folk medicine to treat anxiety, insomnia, and stomachache, among other afflictions associated to pain. *A. mexicana* has been divided in two subspecies: *A. mexicana* ssp. *mexicana* (Toronjil morado) and *A. mexicana* ssp. *xolocotziana* (Toronjil blanco), both of which are used in traditional medicine to alleviate visceral pain; however it was found that only a polar extract from ssp. *mexicana* produced spasmolytic-like effects [111]. This antinociceptive response was demonstrated in a significant and dose-dependent manner in an in vitro study, where ursolic acid and acacetin evaluated by the enteral and parenteral route of administration were both partial responsible constituents [112]. In contrast, ssp. *xolocotziana* was associated with a spasmogenic response. The spasmolytic effects of *A. mexicana* were related to an activation of nicotinic receptors, prostaglandins and calcium channels, but not nitric oxide mechanisms [113]. It is well-known that the abundant presence of certain constituents depends on the manner of preparation of the vegetal material [111]. In the methanol extracts of *A. mexicana*, the abundant presence of flavonoids such as acacetin and tilianin has been found [113]. Extracts from different polarities have been compared in different experimental models of pain, such as the writhing test in mice, the formalin and plantar tests in mice or rats, as well as the pain-induced functional impairment assay in rats (a gouty arthritis pain model) to demonstrate significant and dose-dependent antinociceptive responses. The effect was more evident in the less polar extracts in part due to the presence of ursolic acid [114].This triterpene produced its antinociceptive effect mediated by the presence of cGMP and an additive synergism with 5HT_1A_ receptors, but also produced antagonistic activity towards TRPV1 receptors in neurogenic and inflammatory nociception with an ED_50_ = 44 mg/kg. A lower dosage was required to produce an antinociceptive effect in abdominal pain with an ED50 = 2 mg/kg [115].

Another species from Lamiaceae independent of the *Agastache* genus, but also known as “toronjil” in Mexico is *Clinopodium mexicanum* (Benth.) Govaerts, which is used in Mexican traditional medicine to induce sleep, as well as in a sedative and analgesic remedy with the common name of “Toronjil de Monte”. Its aqueous extract was able to inhibit central nociception using a thermal stimulus in mice supporting its depressant activity, where glavanone glycosides such as neoponcirin, poncirin and isonaringin were involved as bioactive constituents [51].

### 2.7. Secondary Metabolites Identified in Lamiaceae Species with Analgesic and/or Anti-Inflammatory Activities

Given the broad range of known mechanisms for pain transmission, numerous natural compounds of different origins are reported in the literature to directly or indirectly modulate pain transmission to produce analgesic effects. Most of these constituents modulate the release of endogenous analgesic mediators or inhibit algogenic neurotransmitters through pre- or post-synaptic mechanisms at both the central and peripheral levels.

#### 2.7.1. Terpenes

##### Volatile Terpenes

Terpenes are a group of secondary metabolites with a great diversity of chemical structure. This type of compound comprises approximately 90% of the components of the essential oils of aromatic plants [116]. The essential oil of the species of the Lamiaceae is particularly rich in volatile monoterpenes, sesquiterpenes and diterpenes, which are made up of 10, 15 and 20 carbon atoms, respectively [1]. Terpenes are very diverse in both structure and function, but chemically they derive from the polymerization of isoprene; in fact, their classification is based on the number of isoprene units that bind to each other: hemi (one unit), mono (two units), sesqui (three units), di (four units), ses (five units), tetra (eight units), and polyterpenes (*n*-units). Monoterpenes, the main active ingredients in essential oils, consist of two isoprene units that are made up of five carbons joined [117].

Among the monoterpenes, the main reported compounds are α-pinene, β-pinene, 1,8-cineole, menthol, limonene, and γ-terpinene. The monoterpenes commonly present in the Lamiaceae family and whose antinociceptive, anti-inflammatory and/or antioxidant mechanisms of action have been evaluated, as well as their described chemical structure, are listed in Table 2. Anticancer, antimicrobial, antioxidant, antiviral, analgesic and anti-inflammatory activities are attributed to these compounds in various plant species [118]. Regarding the development of analgesics and anti-inflammatories, monoterpenes and sesquiterpenes have become a topic of interest with an increasing number of new patent applications [1,119,120,121]. According to some studies, monoterpenes are promising in the modulation of cytokines due to their lipophilic characteristics which favor their absorption and rapid action [121], and they have been recognized as stimulating an increase in anti-inflammatory cytokines, such as IL-10 [122,123]. Studies of *Hyptis spicigera* Lam. reported the antinociceptive effects of the essential oil because of the presence of α- and β-pinene, and 1,8-cineol associated to the participation of TRPV1, A1 and M8 receptors [58], whereas in the case of the essential oil of *Monarda fistulosa* L., carvacrol, thymol and β-myrcene were characterized as possible compounds responsible for the antinociceptive properties mediated by TRPA1 receptors [124]. The presence of δ-cadinene, α-pinene, myrcene, β-caryophylene, germacrene, and limonene in the essential oil of *Teucrium stocksianum* Boiss. was characterized as a bioactive antinociceptive in the writhing test [125] but also in the formalin test in the antinociceptive activity of *Ocimum* [126].

##### Non-Volatile Terpenes

Non-volatile diterpenes (made up 20 carbon atoms) and triterpenes (made up 30 carbon atoms) are reported as the two main subclasses of components of species in the Lamiaceae family [146]. Seven main types of non-volatile diterpenes have been reported differing in the various arrangements of the 20-carbon atom structure in order to form abietanes, clerodanes, ent-kauranes, iso-pimarans, labdanes and neo-clerodanes [117]. Some of these types are considered chemotaxonomic markers for specific subfamilies or genera [147], although all of them can be indifferently evidenced in all Lamiaceae species.

Much attention has recently been paid to diterpenoids such as marrubiin, from *Marrubium vulgare* L. assayed in experimental models of pain, such as writhing, formalin and hot-plate tests, but also carnosol and carnosic acid [148], which suppress cyclooxygenase (COX)-2, interleukin-1B, and TNF-alfa expression, as well as leukocyte infiltration in inflamed tissues [149,150].

Regarding triterpenes, two main types have been reported: pentacyclic triterpenes and phytoecdysteroids. The former is characterized by having an olean- and ursan-like base structure [151], with oleanolic acid and ursolic acid being the most reported pentacyclic triterpenes in species belonging to this family. Diterpenes and triterpenes within the Lamiaceae have also been reported to produce antinociceptive and/or anti-inflammatory effects. Their chemical structure and mechanism of action explored are listed in Table 3.

#### 2.7.2. Phenolic Compounds

Some species of the Lamiaceae family and their biological activities have been characterized by the presence of phenolic compounds, even if only in minor quantities. Caffeic acid and rosmarinic acid, together with their derivatives, are the most reported phenolic acids in the family [170,171,172]. However, in recent studies, chemotaxonomic markers at the genera level have also been found [121].

In general terms, phenols and polyphenols refer to a group of plant secondary metabolites which carry at least one phenolic ring in their molecule; they are derived from shikimic acid and phenylpropanoid metabolism pathways. A phenolic ring is made up of a hydrophobic aromatic nucleus and a hydrophilic hydroxy group that can be involved in hydrogen bond formation. As redox active compounds, plant phenols can also act as antioxidants or pro-oxidants [173]. The antioxidant activity of phenolic compounds depends on the number of hydroxy substituents, their position and the site of binding on the aromatic ring [174].

All phenolic compounds of the plants in the Lamiaceae share significant antioxidant activity which is attributed to the complete extracts [175]. Antioxidants protect plant cells from damage caused by free radicals that develop with normal cellular metabolism or due to stressful events, such as excessive UV radiation, exposure to soil or air pollutants, and diseases [176]. The antioxidant properties of phenolic compounds can participate in the uptake of reactive oxygen species and reactive nitrogen species (ROS/RNS), inhibiting their formation by suppressing enzymes or metals associated with the production of free radicals, and regulating or defending the plant antioxidant systems [177].

Lipid peroxidation processes which cause damage to fatty acids tend to decrease membrane fluidity and lead to numerous pathological events [178], and could be reduced by phenolic acids in plants due to their capacity to modulate different oxygen species [121,175]. Furthermore, polyphenols have been demonstrated to protect the nervous system against oxidative stress, to such an extent that regular dietary intake of flavonoids has been associated with reduced dementia and delayed onset of Alzheimer’s and Parkinson’s diseases [179]. Polyphenols have been considered potential neuroprotective and direct neuromodulatory agents of the CNS because of their ability to cross the blood-brain barrier [180]. Plants belonging to the genera *Calamintha*, *Lavandula*, *Mentha*, *Melissa*, *Origanum*, *Rosmarinus*, *Salvia*, *Teucrium* or *Thymus* are used for the treatment of various nervous system disorders, mainly thanks to the presence of polyphenols, particularly rosmarinic acid [78].

Phenolic compounds present in the Lamiaceae and their corresponding structures, which have shown antioxidant, anti-inflammatory and/or antinociceptive pharmacological activity, related to the evaluated mechanisms of action, are shown in Table 4.

##### Phenolic Acids

Phenolic acids have been characterized as being responsible for the biological activity of several Lamiaceae species, such as antioxidant, anti-inflammatory and antinociceptive activities, exploring not only their pharmacological activity but also the possible mechanism of action involved (Table 4). As redox active compounds, phenolic acids can also act as antioxidants or pro-oxidants [173]. The antioxidant activity of phenolic compounds depends on the number of hydroxy substituents, their position and the site of binding on the aromatic ring [174]. Caffeic acid and rosmarinic acid and their derivatives have been part of the most common phenolic acids described [170,171,172]. Nevertheless, recent studies have found and established chemotaxonomic markers at the genera level [121].

Rosmarinic acid is among the main phenolic compounds contained in the tissues of various plant species belonging to this family. The genotype of the plant, but also physiological or environmental factors, such as phenology, climate, growth technique and stress conditions, strongly influence the amounts of phenolic compounds in the plant [121,196].

High levels of rosmarinic acid are commonly found only within the Nepetoideae subfamily. In the genus *Stachys*, in the *Lamioideae* subfamily [197], several species belonging to the Lamiaceae can accumulate high levels of different phenolic compounds, such as phenolic acids, flavonoids, or phenolic terpenes. Only in the Lamiaceae family are some phenolic compounds present, such as carnosic acid, which prevents the oxidative damage of the chloroplast and shows highly antioxidant properties in vitro [198]. Another phenolic acid exclusive to Lamiaceae species is clerodendranoic acid, which was found in *Clerodendranthus spicatus* (Thunb.) C.Y. Wu ex H.W. Li [199].

##### Flavonoids

A few compounds from flavonoid nature in Lamiaceae with antinociceptive and anti-inflammatory activities have been reported. This is the case of a standardized mixture of baicaline and catequine from *Scutellaria baicalensis* Georgi, which was evaluated using the writhing, formalin and carragennan tests in rodents [200]. Salvigenina was isolated from *S. officinalis* as the responsible flavonoid for the antinociceptive effects in writhing, hot-plate and carragennan tests [201]. Other flavonoids including pedalitin from *S. amarissima* have been explored in writhing test in mice [102], while tilianin and acacetin were identified in *Agastache mexicana* and reported in several experimental models of acute and chronic pain [113]. Examples of these flavonoids commonly present in species of the Lamiaceae family with antioxidant, antinociceptive and anti-inflammatory activity are listed in Table 5.

## 3. Materials and Methods

### Literature Survey Databases

This literature review was carried out based on an electronic search in the Sciencedirect, Pubmed and Springer link databases in 2020. The keywords used were “Lamiaceae” “antinociceptive” “pain”, “analgesic” and “anti-inflammatory”. Almost 1200 articles were found, and after an extensive survey, 217 articles were selected, which described the antinociceptive and/or anti-inflammatory potential of natural compounds to relieve pain.

## 4. Conclusions

The chemical characteristics and the pharmacological properties of the Lamiaceae constituents are of interest to researchers, laboratories, and pharmaceutical companies. During the last few decades, the mechanisms of action of the different secondary metabolites of the Lamiaceae family have been broadly investigated by means of in vivo and in vitro assays to confirm their participation in the modulation of pain and in the cascade of inflammation mediators. This work summarizes part of the reported scientific knowledge regarding the secondary metabolites of some specific Mexican species of the Lamiaceae that have shown activity for pain relief, highlighting the participation of terpenes, flavonoids, and phenolic acids as potential alternatives for new drug therapies. As a result of this review, it is important to mention that few studies have been reported regarding Mexican genera of this family; for example, *Calosphace* is one of the largest subgenera of *Salvia* in all the world, mainly found in Mexico, but it has barely been investigated regarding its potential biological activities and their bioactive constituents. The scientific evidence regarding the different bioactive constituents found in species of the Lamiaceae family demonstrates that several species of this family require further investigation in preclinical studies, but also in controlled clinical trials to evaluate the efficacy and safety of these natural products to support their therapeutic potential in pain relief and/or inflammation, along with other health conditions.

## Figures and Tables

**Figure 1 molecules-26-07632-f001:**
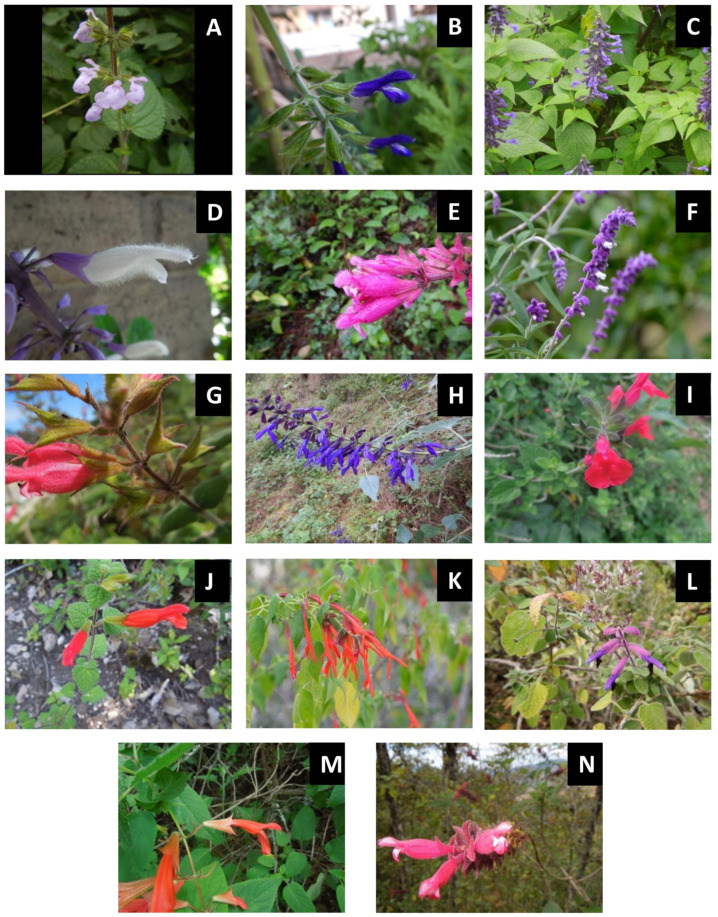
Photographs of examples of Lamiaceae from Mexican Salvias (*Calosphace* subgenus). (**A**): S. *circinnata* Cav., (**B**): *S. calderoniae* Bedolla & Zamudio, (**C**): *S. concolor* Lamb. ex Benth., (**D**): *S. divinorum* Epling & Játiva, (**E**): *S. involucrate* Cav., (**F**): *S. leucantha* Cav., (**G**)*: S. karwinskii* Benth., (**H**): *S. mexicana* L. (**I**): *S. microphylla* Kunth, (**J**): *S. oaxacana* Fernald, (**K**): *S. pubescens* Benth., (**L**): *S. semiartrata,* (**M**): *S. tilantongensis* J.G. González & Aguilar-Sant. and (**N**): *S*. *wagneriana* Zucc.

**Figure 2 molecules-26-07632-f002:**
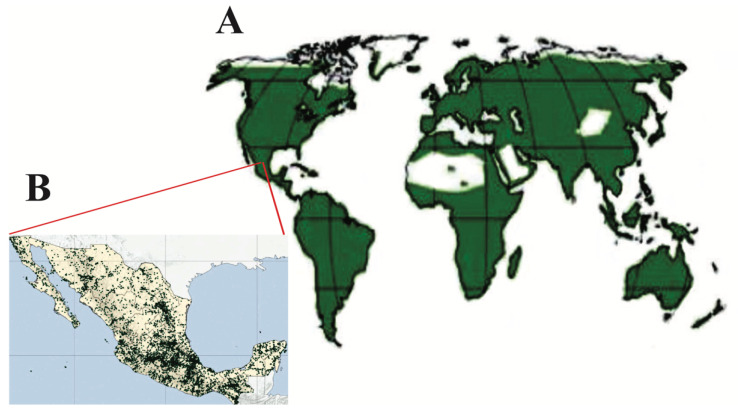
(**A**) Worldwide distribution of plants of the Lamiaceae family [1] and (**B**) in Mexico (Modified from Martínez-Gordillo et al. [28]).

**Figure 3 molecules-26-07632-f003:**
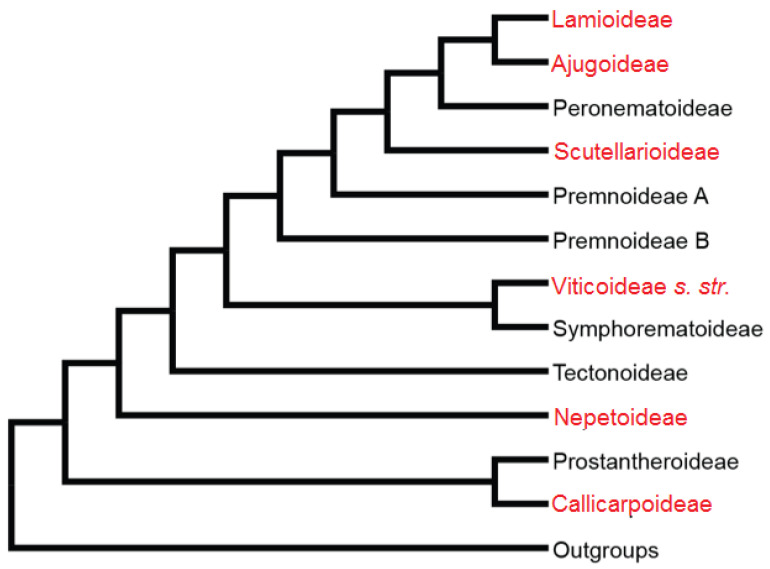
Cladogram of Lamiaceae showing the monophyletic clades, where the different subfamilies are recognized. Subfamilies recognized by Olmstead: *Ajugoideae*, *Lamioideae*, *Nepetoideae*, *Prostantheroideae*, *Scutellarioideae*, *Symphorematoideae*, *Tectonoideae*, *Callicarpoideae* and *Viticoideae* (Modified from Li et al. [37]).

**Table 1 molecules-26-07632-t001:** Lamiaceae species used in traditional medicine for pain, inflammation treatment and/or as antioxidants.

Scientific Name	Medical Properties			Used Plant Organs	Preparation
	Analgesic	Anti- Inflammatory	Antioxidant		
*Clinopodium vulgare* L. [50]		X		Aerial parts	Hydroalcoholic extract
*C. mexicanun* (Benth.) Govaerts [51]	X			Leaves	Organic extracts
*Eremostachys laciniata* (L.) Bunge. [52]			X	Aerial parts	Hydrodistillation, Methanol extract
*Glechoma longituba* Kupr. [53]			X	Aerial parts	Infusion
*Hedeoma drummondii* Benth. [54]			X	Aerial parts	Maceration
*H. multiflorum* Benth. [55]			X	Aerial parts	Infusion
*Holmskioldia sanguinea* Retz. [56]		X		Leaves	Methanol extract
*Hyptis suaveolens* (L.) Poit. [57]	X			Aerial parts	Hydroalcoholic extract
*H. spicigera* Lam. [58]	X	X		Aerial parts	Hydrodistillation
*Lamium**álbum* L. [59]		X	X	Aerial parts	Hydroalcoholic extract
*Lavandula angustifolia* Mill. [60,61]	X	X	X	Leaves, aerial parts	Hydrodistillation, Ethanol extract
*Leonorus**cardiaca* L. [62]	X			Aerial parts	Hydroalcoholic extract
*Leonotis leonorus* L. [63]		X	X	Aerial parts	Organic extracts
*Leucas**aspera* Link [64]	X		X	Roots	Maceration
*Marrubium**vulgare*. L. [65]			X	Leaves, aerial parts	Tincture, Organic extracts
*Mentha**piperita* L. [66]		X	X	Leaves	Hydrodistillation
*M. suaveolens* Ehrh. [67]	X	X	X	Aerial parts	Methanol extract
*Ocimum americanum* L. [68]			X	Aerial parts	Methanol extract
*O. basilicum* L. [69]	X	X		Aerial parts	Hydrodistillation
*Phlomis**purpurea* L. [70]	X	X		Aerial parts	Methanol extract
*P. nissolii* L. [71]			X	Leaves	Decoction
*Premna herbacea* Roxb. [72]	X	X		Roots	Ethanol extract
*P. integrifolia* L. [73]		X		Roots	Organic and aqueous extracts
*Prunella**vulgaris* L. [74]	X		X	Inflorescence	Ethanol extract
*Rosmarinus officinalis* L. [75,76]	X	X	X	Aerial parts, leaves	Maceration, Methanol extract
*Salvia officinalis* L. [77,78,79]	X	X	X	Aerial parts, Leaves	Infusion, Decoction, Hydroalcoholic extract
*S. hispanica* L. [80]			X	Aerial parts	Organic extracts
*Scutellaria indica* L. [81]		X		Aerial parts	Organic extracts
*S. baicalensis* Georgi. [82]		X	X	Aerial parts, Roots	Aqueous extract, Organic extract
*Sideritis bilgeriana* P.H. Davis [83]	X	X	X	Aerial parts	Maceration
*S. congesta* P.H. Davis & Hub.-Mor. [84]			X	Aerial parts	Maceration
*Stachys**byzantina* C. Koch. [85]	X	X		Aerial parts	Organic extracts
*S. inflata* Benth. [86]		X		Aerial parts	Hydroalcoholic extract
*Thymus**serpyllum* L. [87]			X	Aerial parts	Hydrodistillation
*T.**vulgaris* L. [88]			X	Leaves	Hydrodistillation
*Vitex**agnus-castus* L. [89]		X		Leaves	Methanol extract
*V. megapotamica* Cham. [90]	X	X		Leaves	Hydroalcoholic extract

The scientific names were confirmed in The Plant List. Available online: http://www.theplantlist.org/ (Accessed on 2 December 2021).

**Table 2 molecules-26-07632-t002:** Monoterpenes and sesquiterpenes present in Lamiaceae with biological activity and their molecular targets explored in pain and inflammation.

Compound	Structure	Mechanism of Action	References
β-pinene	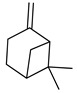	Decreased expression of IL-6, TNF-α, NO, iNOS and COX-2.Down-regulation of MAPKs phosphorylation and the NF-κB signaling pathway	[127]
Inhibition of COX-2 enzyme expression	[128]
Limonene	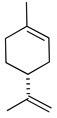	Reduction in leukocyte infiltration and TNF-α levels.	[129]
Decreased production of NO, PGE2 and Pro-inflammatory cytokines	[130]
Linalool	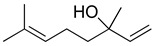	Inhibition of pro-inflammatory interleukins and modulation of NMDA glutamatergic receptor.	[131]
Reduction in oxidative stress and inflammation (NF-kB).	[132]
Activation of opioid and muscarinic receptors	[133]
Myrcene	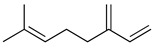	Activation of opioid receptors and presynaptic α2 adrenoreceptor.	[134]
Inhibition of IL-1β-induced NO productionIncreased expression of TIMP-1 and TIMP-3.	[135]
p-cymene	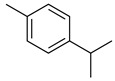	Reduced the production of pro-inflammatory cytokine TNF-α, the migration of leukocytes, and the release of NO. Activation of opioid receptors.	[136]
Reduced the calcium current density.	[137]
Thymol	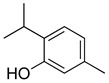	Voltage-operated sodium channel blocker	[138]
TRPA1 channel presynaptic activation	[139]
Carvacrol	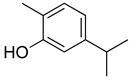	Inhibition of expression TNF-α, IL-1β, and IL-6Reduced the expression of NF-kB	[140]
Modulation of opioid, vanilloid and glutamate systems	[141]
α-humulene	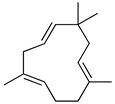	Inhibition of pro-inflammatory cytokines (TNF-α and IL-1β) and PGE2 generation.Decreased expression of iNOS and COX-2.Inhibition of Il-5, CCL11 and LTB4 levels and P-selectin expression.	[142][143]
β-caryophyllene	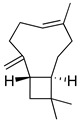	Cannabinoid receptor type 2 agonist.Attenuation of Substance P and cytokines such as IL-1β, TNF-α, and IL-6.	[144]
Agonist to opioid, benzodiazepine, 5HT1A receptors and NO.	[145]

Abbreviations: COX-2: Cyclooxygenase-2; IL-: Interleukin-; iNOS: Inducible nitric oxide synthase; LTB4: Leukotriene B4; MAPKs: Mitogen-activated protein kinase; NF-kB: Nuclear factor kappa-light-chain-enhancer of activated B cells; NMDA: N-methyl-D-aspartate receptor; NO: Nitric oxide; PGE2: Prostaglandin E2; TIMP-: Tissue inhibitors of metalloproteinase-; TNF-α: Tumor necrosis factor-alpha; TRPA1: Transient receptor potential cation channel, subfamily A, member 1.

**Table 3 molecules-26-07632-t003:** Diterpenes and triterpenes present in Lamiaceae with biological activity and their molecular targets explored in pain and inflammation.

Compound	Structure	Mechanism of Action	References
Tormentic acid	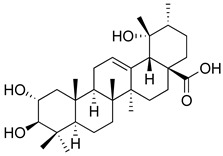	Inhibition of NF-kB signaling pathway and prevents the expressions of iNOS, COX-2, and TNF-α.	[152]
		Increased activity of Superoxide dismutase, glutathione peroxidase and catalase.	[153]
Andalusol	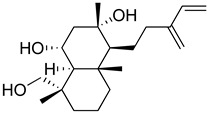	Inhibition of histamine	[154]
		Inhibition of iNOS expression by inactivation of NF-kB	[155]
Tanshinone IIA	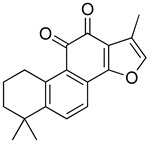	TLR2/NF-kB signaling pathway blocker	[156]
Salvinorin A	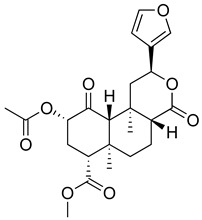	KOR agonist.	[157]
		Inhibition of dopamine overflow mediated by KOR.	[158]
α-amyrenone	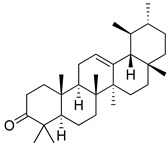	PKC and PKA activity blocker.	[159]
		Antioxidant activity.	[160]
β-amyrenone	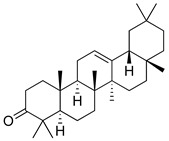	Decreased levels of TNF-α and caspase 3 Reduction in oxidative stress.	[161]
Ursolic acid	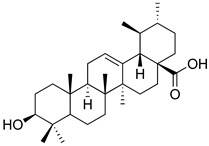	NO, PGE2 inhibitor.	[162]
		TRPV1 antagonist. Modulator of cGMP and serotonergic system.	[115]
Carnosol	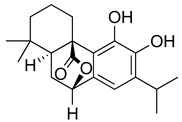	Suppression of iNOS by down-regulation of NF-kB.	[163]
		Suppression of PGE2 synthesis by the inhibition of mPGES-1.	[164]
		Inhibition of the induction of COX-2 by blocking PKC signaling and thereby the binding of AP-1 to the CRE of the COX-2 promoter.	[165]
Oleanolic acid	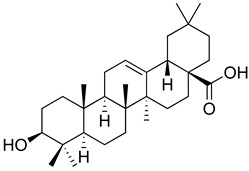	Opioid agonist. NO inhibitor. Activation of ATP-gated K^+^ channels.	[166]
		Opioid and 5-HT agonist.	[167]
Betulinic acid	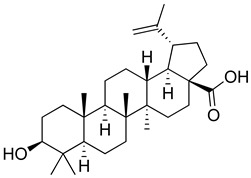	Reduction in TNF-α production. Increase in IL-10 production.	[168]
		Reduction in the levels of COX-2, NO, TNF-α and IL-1β. Inhibition of MDA level via increasing the activities of SOD, GPx, GRd.	[169]

Abbreviations: 5HT: 5-hydroxytryptamine; 5HT_1A:_ Serotonin 1A receptor; AP-1: activator protein 1; ATP: Adenosine triphosphate; CCL11: C-C motif chemokine 11; cGMP: Cyclic guanosine monophosphate; COX-2: Cyclooxygenase-2; CRE: cyclic AMP response element; GPx: glutathione peroxidase; GRd: glutathione reductase; IL-: Interleukin-; iNOS: Inducible nitric oxide synthase; KOR: κ-opioid receptor; LTB4: Leukotriene B4; MDA: Malondialdehyde; mPGES-1: Microsomal prostaglandin E synthase-1; NF-kB: Nuclear factor kappa-light-chain-enhancer of activated B cells; NO: Nitric oxide; PGE2: Prostaglandin E2; PKA: protein kinase A; PKC: protein kinase C; SOD: Superoxide dismutase; TLR2: Toll-like receptor 2; TNF-α: Tumor necrosis factor-alpha; TRPV1: Transient receptor potential cation channel subfamily V member 1.

**Table 4 molecules-26-07632-t004:** Phenolic acids commonly found in Lamiaceae and their molecular targets explored in pain and inflammation.

Compound	Structure	Mechanisms of Action	References
Rosmarinic acid	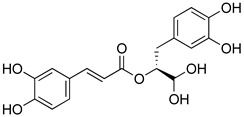	Antioxidant activity.	[75]
		Suppression of TNF-α, iNOs, apoptotic factors (Bax, caspases 3 and 9), Iba-1, TLR4 and GFAP levels.	[181]
Gallic acid	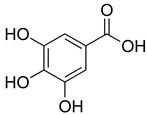	ERK-Nrf2-Keap1-mediated antioxidant activity.	[182]
		Reduction in TBARS, total calcium, TNF-α, superoxide anion, and MPO activity levels; and decreased GSH level.	[183]
		TRPA1 antagonist.	[184]
Chlorogenic acid	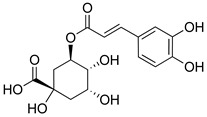	Inhibition of CD80/86 and Th1 cytokines.	[185]
		GABA_A_ receptor agonist.	[186]
		Inhibition of NF-kB and JNK/AP-1 signaling pathways.	[187]
Vanillin	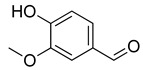	Inhibition of protein and lipid oxidation processes. Increased activity of GSH, SOD, catalase.Suppresses the expression of TNF-α, IL-6, IL-1β and plasma AST and ALT enzymes.	[188]
		α_2_-adrenergic and opioid receptor agonist	[189]
Caffeic acid	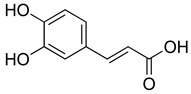	Reduction in the IκBα degradation and p65 phosphorylation in the NF-kB pathway.	[190]
		Inhibition of MPO, MDA and nitrite generation.	[191]
Vanillic acid	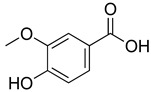	α_2_-adrenoceptor agonist.5HT_3_ and 5HT_1_ receptor agonist Interaction with TRPV1, TRPA1 and TRPM8 receptors.	[192]
		Inhibition of oxidative stress, pro-inflammatory cytokine production, and NF-kB activation.	[193]
Ferulic acid	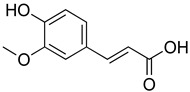	The level/activity of elastase, lysosomal enzymes, nitric oxide, lipid peroxidation, and pro-inflammatory cytokines (TNF-α and IL-1β); and the mRNA expression of NLRP3 inflammasomes, caspase-1, pro-inflammatory cytokines, and NF-kB p65 were decreased.	[194]
		Inhibition of xanthine oxidase and COX-2 enzyme.	[195]

Abbreviations: 5HT: Serotonin; ALT: Alanine aminotransferase; AP-1: Activator protein 1; AST: Aspartate aminotransferase; Bax: Bcl-2-associated X protein; CD80/86: CD28 receptor binds to the B7; COX-2: Cyclooxygenase-2; ERK: Extracellular-signal-regulated kinase; GABA_A_: γ-aminobutyric acid type A receptor; GFAP: Glial fibrillary acidic protein; GSH: Glutathione; Iba-1: Ionized calcium-binding adapter molecule 1; IL-: Interleukin-; iNOS: Inducible nitric oxide synthase; IkBα: Nuclear factor of kappa light polypeptide gene enhancer in B-cells inhibitor, alpha; JNK: c-Jun N-terminal kinase; Keap1: Kelch-like ECH-associated protein 1; MDA: Malondialdehyde; MPO: Myeloperoxidase; mRNA: Messenger Ribonucleic acid; NF-kB: Nuclear factor kappa-light-chain-enhancer of activated B cells; NLRP3: Family pyrin domain containing 3; NRf2: nuclear factor erythroid 2–related factor 2; p65: Nuclear factor NF-kappa-B p65 subunit; SOD: Superoxide dismutase; TBARS: Thiobarbituric acid reactive substances; Th1: T helper type 1; TLR4: Toll like receptor 4; TNF-α: Tumor necrosis factor-alpha; TNF-α: Tumor necrosis factor-alpha; TRPA1: Transient receptor potential ankyrin 1; TRPA1: Transient receptor potential cation channel, subfamily A, member 1; TRPM8: Transient receptor potential cation channel subfamily M (melastatin), member 8; TRPV1: Transient receptor potential cation channel subfamily V member 1.

**Table 5 molecules-26-07632-t005:** Flavonoids commonly present in Lamiaceae and their molecular targets explored in pain and inflammation.

Compound	Structure	Mechanism of Action	References
Pedalitin	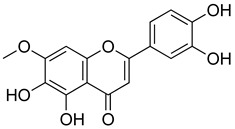	Inhibitory effects against NO, TNF-α and IL-12.	[202]
Rutin	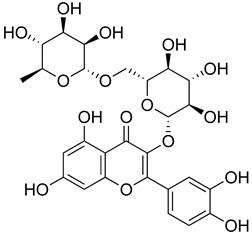	Increased activity of GPx, GRd, CAT, SOD and GSH.	[203]
		Central modulation of the vlPAG descending circuit partly mediated by an opioidergic mechanism.	[106]
		Increased H_2_S level.Modulation of Nrf2 pathway. Caspase 3 and, NF-kB, TNF-α, IL-6 decreased.Increased sensory nerve conduction velocity.	[204]
Apigenin	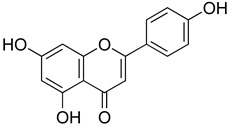	Increased expression levels of Nrf2 and HO-1.Inhibition of TNF-α, IL-1β, IL-6, MPO and MDA content.	[24]
		Inhibition of CD40, TNF-α and IL-6	[205]
Quercetin	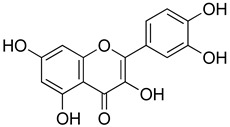	Interaction with *L*-arginine-nitric oxide, serotonin, and GABAergic systems.	[206]
		ROCs and VOCs Blocker Modulation of PGF2α pathway	[207]
		5HT_1A_ agonist	[208]
Luteolin	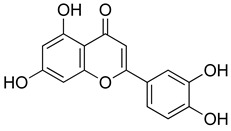	Inhibition of IL-1β, TNF-α and histamine release.	[209]
		Decreased neutrophil infiltration.Inhibition of TNF-α, IL-1β, IL-6.	[210]
		Downregulation of TLR4/TRAF6/NF-kB pathway	[211]
		Inhibition of CD40, TNF-α and IL-6	[205]
Hesperidin	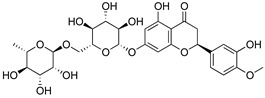	Modulation of D_2_, GABA_A_ and opioid receptors.	[212]
		Agonist of opioid receptors.	[213]
		Modulation of TRPV1 receptor.	[189]
Naringin	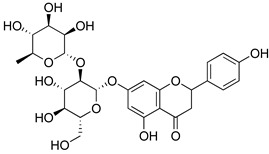	Inhibition of oxido-nitrosative strees, TNF-α, IL’s and NF-kB mRNA levels.	[214]
		Inhibition of PGE2, NO, IL-6 and TNF-α.	[215]
Naringenin	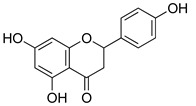	Inhibition of NF-kB and activation of NO-Cyclic GMP-PKG-ATP sensitive K^+^ channel pathway	[216]
		Inhibition of IL-6, TNF-α and NO release, by interfering MAPK signal pathway and suppressing the activation of NF-kB.	[217]

Abbreviations: 5HT_1A_: Serotonin 1A receptor; ATP: Adenosine triphosphate; CAT: Catalase; CD40: Cluster of differentiation 40; GABA: γ aminobutyric acid; GMP: Cyclic guanosine monophosphate; GPx: glutathione peroxidase; GRd: glutathione; reductase; GSH: Glutathione; H_2_S: Hydrogen sulfide; HO-1: Heme oxygenase-1; IL-: Interleukin-; MAPK: Mitogen-activated protein kinase; MDA: Malondialdehyde; MPO: Myeloperoxidase; mRNA: Messenger Ribonucleic acid; NF-kB: Nuclear factor kappa-light-chain-enhancer of activated B cells; NF-kB: Nuclear factor kappa-light-chain-enhancer of activated B cells; NO: Nitric oxide; Nrf2: Nuclear factor erythroid 2–related factor 2; PGE2: Prostaglandin E2; PGF2α: Prostaglandin F2α: PKG: cGMP-dependent protein kinase ROCs: Receptor-operated channels; SOD: Superoxide dismutase; TLR4: Toll-like receptor 4; TNF-α: Tumor necrosis factor-alpha; TRAF6: TNF receptor associated factor 6; TRPV1: Transient receptor potential cation channel subfamily V member 1; VOCs: Voltage-operated channels.

## Data Availability

The data presented in this study are available on request from the corresponding author.

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
