# Peer review of "Lamiaceae in Mexican Species, a Great but Scarcely Explored Source of Secondary Metabolites with Potential Pharmacological Effects in Pain Relief"

_molecules, 2021, doi:10.3390/molecules26247632_

Round 1

Reviewer 1 Report

The manuscript submitted for re-evaluation was significantly improved by the authors. Many changes have been made that have been highlighted as those not highlighted by the authors with the appropriate color. Nevertheless, I believe that in the abstract and in the introduction there is no indication of the novelty of this study. Despite the great effort that the authors put into modifying the entire text, it is still necessary to demonstrate what distinguishes the presented study from previous publications. In addition, it would be advisable for authors to liven up their article a bit. The introduction of figures or photographs of the pots in question would be a good distinguishing feature, which would also encourage potential readers to read this study.

Author Response

We gratefully appreciate all the suggestions and comments of the reviewer, all of them were considered for improvement of this manuscript. We have emphasized the objective and highlights of this review in the abstract and introduction. In addition, it was included a new figure with photographs to exemplify the presence of some Mexican species from Calosphaceae subgenus of this family to better catch the interest of lectors. 

Reviewer 2 Report

Comments have been modified and can be published

Author Response

We gratefully appreciate all the suggestions and comments of the reviewer to improve our manuscript to be publishable in molecules.

This manuscript is a resubmission of an earlier submission. The following is a list of the peer review reports and author responses from that submission.

Round 1

Reviewer 1 Report

Introduction
- New resources to be used
-Introduction is too short to upgrade
- The purpose of the study should be added to the end of the introduction
Results-
-
In Table 1, the names of the plants should be mentioned as species, not genus
In Table 2, many compounds from the Lamiaceae family do not have the name of the genus or species. Why compare with other families? It was better to mention the name of the species along with the composition
The same in Table 3

Reviewer 2 Report

The manuscript, entitled: Lamiaceae in Mexican species as a source of secondary metabolites with potential pharmacological effects in pain relief ", is another work on the possibility of using natural substances in pharmacology as a potential source of therapeutic agents.
Unfortunately, the reviewed article does not bring new content to the existing state of knowledge. In its form and content, it repeats many previous reports. There is no indication of any novelties in relation to the reviews presented so far in the field of the subject. Both the system of the substances and activities in question, as well as tables, which, like many others, contain the names of compounds, structural formula, species and / or the family in which the active substance in question has been identified or its presence has been confirmed, and the mechanism of action.
All this means that in my best opinion and caring for the quality of the journal, I cannot recommend this article for publication.